# Modeling Multivariate Distributions of Lipid Panel Biomarkers for Reference Interval Estimation and Comorbidity Analysis

**DOI:** 10.3390/healthcare13192499

**Published:** 2025-10-01

**Authors:** Julian Velev, Luis Velázquez-Sosa, Jack Lebien, Heeralal Janwa, Abiel Roche-Lima

**Affiliations:** 1Department of Physics, University of Puerto Rico, Puerto Rico, PR 00925-2537, USA; 2Abartys Health, San Juan, PR 00907-3913, USA; jlebien@abartyshealth.com; 3Center for Collaborative Research in Health Disparities, RCMI Program, Medical Science Campus, University of Puerto Rico, San Juan, PR 00936-5067, USA; luisfernandojavier.velazquez@upr.edu (L.V.-S.); abiel.roche@upr.edu (A.R.-L.); 4Department of Mathematics, University of Puerto Rico, Puerto Rico, PR 00925-2537, USA; heeralal.janwa@upr.edu

**Keywords:** reference intervals, clinical laboratory data, Gaussian mixture models

## Abstract

**Background/Objectives**: Laboratory tests are a cornerstone of modern medicine, and their interpretation depends on reference intervals (RIs) that define expected values in healthy populations. Standard RIs are obtained in cohort studies that are costly and time-consuming and typically do not account for demographic factors such as age, sex, and ethnicity that strongly influence biomarker distributions. This study establishes a data-driven approach for deriving RIs directly from routinely collected laboratory results. **Methods**: Multidimensional joint distributions of lipid biomarkers were estimated from large-scale real-world laboratory data from the Puerto Rican population using a Gaussian Mixture Model (GMM). GMM and additional statistical analyses were used to enable separation of healthy and pathological subpopulations and exclude the influence of comorbidities all without the use of diagnostic codes. Selective mortality patterns were examined to explain counterintuitive age trends in lipid values while comorbidity implication networks were constructed to characterize interdependencies between conditions. **Results**: The approach yielded sex- and age-stratified RIs for lipid panel biomarkers estimated from the inferred distributions (total cholesterol, LDL, HDL, triglycerides). Apparent improvements in biomarker profiles after midlife were explained by selective survival. Comorbidities exerted pronounced effects on the 95% ranges, with their broader influence captured through network analysis. Beyond fixed limits, the method yields full distributions, allowing each individual result to be mapped to a percentile and interpreted as a continuous measure of risk. **Conclusions**: Population-specific and sex- and age-segmented RIs can be derived from real-world laboratory data without recruiting healthy cohorts. Incorporating selective mortality effects and comorbidity networks provides additional insight into population health dynamics.

## 1. Introduction

Cardiovascular disease (CVD) remains the leading cause of death in the United States, responsible for roughly one in every five deaths, and has held this position since 1950 [1,2]. Among the spectrum of CVDs, coronary artery disease (CAD) is the most prevalent, claiming over 370,000 lives in 2022, and affecting approximately 5% of adults aged 20 and older [2,3]. Established risk factors include non-modifiable traits such as age, sex, race, and genetics, as well as modifiable lifestyle factors—notably hypertension, hyperlipidemia, diabetes, obesity, smoking, poor diet, and physical inactivity [4]. While genetic predisposition remains significant, the growing obesity epidemic and its metabolic consequences—like diabetes and dyslipidemia—have increasingly driven the rise in cardiac conditions, particularly in developed nations.

Because lipid abnormalities are central to the development and progression of cardiovascular disease, the lipid panel (LP) has become the main laboratory tool for monitoring cardiovascular health and assessing risk [5,6,7]. It includes total cholesterol (CHOL), triglycerides (TRIG), high-density lipoprotein cholesterol (HDL), low-density lipoprotein cholesterol (LDL), and very-low-density lipoprotein (VLDL). These biomarkers represent different classes of lipoproteins—complexes of lipids and proteins that transport cholesterol and triglycerides through the bloodstream. HDL is termed “good cholesterol” because it helps remove excess cholesterol from tissues and plaques, whereas LDL is labeled “bad cholesterol” since elevated levels promote atherosclerosis and increase CVD risk [8]. Triglycerides, the main form of stored fat, are linked to insulin resistance, metabolic syndrome, and higher CVD risk, particularly when HDL is low [9].

Comorbidities have a significant impact on biomarker levels, and understanding their influence is essential for correctly interpreting LP results. In this study, we focus on two highly prevalent conditions—diabetes mellitus (DM) and chronic kidney disease (CKD)—that strongly affect cardiovascular outcomes [7]. Diabetes is a major risk factor for atherosclerotic cardiovascular disease through its effects on insulin resistance, dyslipidemia, and systemic inflammation [10]. CKD is likewise accompanied by dyslipidemia, typically characterized by elevated triglycerides and reduced HDL concentrations [7,11]. Their two key diagnostic markers—glycated hemoglobin (A1C) for diabetes and serum creatinine (CREA) for renal function—are included in the comprehensive metabolic panel (CMP) [5]. Since LP and CMP are often ordered together in routine general health screening, ample data are available for assessing how comorbidities influence lipid biomarkers.

Interpretation of these biomarkers depends on reference intervals (RIs), which define the expected range of values in healthy populations [12,13,14,15]. This term is preferred over “normal range” because it emphasizes comparison with a defined reference group rather than implying an absolute standard of health [14,16]. By convention, RIs correspond to the central 95% of the distribution in a healthy population [13]. Their use has been standardized by the International Federation of Clinical Chemistry (IFCC) and the Clinical and Laboratory Standards Institute (CLSI), which have issued detailed guidelines for their estimation and periodic updating [17].

Standard RIs are established through cohort studies, which require at least 120 healthy individuals for each analyte [17]. This approach is time-consuming and costly, making it difficult to update RIs regularly. As a result, many published intervals are outdated and lack stratification by sex, age, or ethnicity—factors known to strongly influence biomarker distributions [18,19,20,21,22,23]. These limitations can be addressed by indirect methods, which infer RIs from secondary use of data sources not originally collected for this purpose, such as clinical laboratory results [24,25,26].

Building on these advances, we have previously proposed indirect methods to estimate RIs from real-world clinical laboratory data originally collected for diagnostic purposes [27,28,29]. These approaches assume that each biomarker distribution consists of a dominant “healthy” component with pathological results superimposed [30]. Using Gaussian mixture models (GMM), we successfully separated these contributions and derived RIs for biomarkers related to CKD [27] and chronic liver disease (CLD) [29] in the Puerto Rican (PR) population.

In the present work, we extend these studies in two important ways. First, methodologically, we advance the GMM framework to multivariate data, using conditional probability distributions with comorbidity markers to better exclude pathological values without relying on diagnostic codes. This approach not only produces RIs that are more representative of the healthy population but also reveals how comorbidities systematically distort biomarker distributions. Second, epidemiologically, we leverage a very large dataset of laboratory results from the PR population to derive age- and sex-specific RIs at single-year resolution for key CVD-related biomarkers, providing the first large-scale characterization of this kind. In doing so, we also identify population-level patterns such as selective mortality, which help explain counterintuitive improvements in biomarker values at older ages, and we construct comorbidity implication networks to capture interdependencies among conditions and their impact on lipid biomarkers.

## 2. Materials and Methods

### 2.1. Data

In this study, data were obtained from the clinical results datalake of Abartys Health, a clinical laboratory data processor headquartered in San Juan, PR, which aggregates laboratory test results from hundreds of laboratories across the island. All data were de-identified by removing personally identifiable information (e.g., names, dates of birth, addresses); unique patients were assigned non-descriptive identifiers, with only age and sex retained as demographic information. In accordance with US CFR 46.104(d), analysis of de-identified results does not require patient consent. The study protocol was reviewed and approved by the University of Puerto Rico—Medical Sciences IRB (Ref. 2301072914).

After extracting the data for each measure, results are coarse-grained by month: multiple readings for the same individual within a given month are averaged. Most patients have only a single monthly reading, but a small subset—likely hospitalized patients—show multiple results. In such cases, coarse-graining helps reduce bias toward more severely ill individuals. Finally, the datasets for all measures are merged by patient ID and the corresponding year and month of testing.

The dataset covers LP results [5] from 2019 to 2024. Because LP and CMP are routinely prescribed as screening tests during medical visits, the resulting data for these biomarkers is extensive, as summarized in Table 1.

From 2019 to 2024, the dataset contains over 4.3 million LP results covering more than 1.3 million unique individuals, with a comparable number of records for the renal panel. About two million glycemic tests are available in the data. The sex distribution across these datasets is fairly consistent at approximately 42% male and 58% female. When restricted to lipid biomarkers tests performed within the same time window and alongside contemporary glycemic and renal tests, the joint dataset includes about 1.8 million results representing roughly 0.7 million individuals (Table 1). Additional details on the data collection and processing are given in the Appendix A.

### 2.2. Methods

As described in our previous work, GMM was used to analyze the data distribution [27,29].

#### 2.2.1. Transformation

The quantitative biomarkers considered in this study are strictly positive, as they represent concentrations, counts, or ratios. Their empirical distributions are typically either approximately normal or lognormal, depending on the underlying physiology. Many biomarkers are subject to homeostatic regulation, which constrains their values within a narrow physiological range around an equilibrium. Deviations from this equilibrium arise due to differences in body size, metabolic rate, and other natural variations. In such cases, a normal distribution provides an adequate description. Conversely, biomarkers such as those in the lipid panel often display multiplicative variability, leading to skewed distributions that are more appropriately modeled as lognormal. To accommodate this, we apply logarithmic transformations to skewed variables prior to further analysis, thereby stabilizing the variance and producing approximately symmetric distributions suitable for multivariate Gaussian modeling.

Each analyte i have an invertible, monotone transform $T_{i}$ (e.g., $T_{i}\left(x\right)=\log x$ or identity). The model is fit in the transformed space $Z_{i}=T_{i}\left(X_{i}\right)$ where $Z∼N\left(\mu ,\Sigma \right)$ are normally distributed. All conditioning values b (and any bounds used elsewhere) are first mapped to Gaussian space via $b_{i}^{\left(G\right)}=T_{i}\left(b_{i}\right)$. After sampling or computing percentiles/ellipsoids in Gaussian space, results are mapped back analyte-wise by $X_{i}=T_{i}^{-1}\left(Z_{i}\right)$.

#### 2.2.2. Joint Distribution

After log-transforming skewed biomarkers, the resulting data can be modeled using Gaussian mixtures. We assume that the observed population consists of a dominant distribution corresponding to physiologically healthy individuals, with pathological results appearing as secondary modes superimposed on this background. GMM provides a natural framework to disentangle these contributions and identify the principal component associated with health [31].

Formally, the probability distribution of an d-dimensional biomarker vector $X=\left(X_{1},X_{2},\ldots ,X_{d}\right)\in R^{d}$ Gaussian space (after any feature-wise transforms), modeled as a convex combination of multivariate Gaussian densities(1)$$p\left(X\right)=\sum_{k=1}^{K}w_{k}N\left(X\;|\;\mu_{k},\Sigma_{k}\right)$$
where $w_{k}$ are non-negative mixture weights satisfying $\sum_{k=1}^{K}w_{k}=1$, and $\mu_{k}\in R^{d}$ is the mean vector, and $\Sigma_{k}\in R^{d\times d}≻0$ denote the positive definite covariance matrix of the k-th Gaussian component, respectively.

To estimate these parameters from data, we employ a Bayesian Gaussian Mixture Model (BGMM), which performs variational inference rather than standard maximum likelihood estimation. A key advantage of BGMM is its adaptive treatment of the number of active mixture components: instead of fixing K, a Dirichlet process prior allows redundant components to be suppressed, yielding a parsimonious representation. In practice, we implement this approach using the scikit-learn BGMM algorithm [32,33]. The healthy reference distribution is then defined as the principal Gaussian component, i.e., the mixture element with the largest weight $w_{k}$, which captures the central tendency of the majority population.

#### 2.2.3. Marginal and Conditional Distributions

Let $X=\left(X_{1},\ldots ,X_{d}\right)∼N\left(\mu ,\Sigma \right)$ be a d-dimensional Gaussian random vector.

Marginals

For a subset of indices A ⊂{1,…,d}, the marginal distribution of the subvector $X_{A}$ is itself Gaussian, with parameters obtained by restricting the mean and covariance:(2)$$X_{A}∼N\left(\mu_{A},\;\Sigma_{AA}\right)$$
where $\mu_{A}$ and $\Sigma_{AA}$ denote the sub-vector and sub-matrix of μ and Σ, respectively.

In particular, for a single analyte $X_{i}$, the univariate distribution is obtained as the marginal with A = {i}. Marginals reduce the dimensionality of the distribution by integrating out irrelevant axes (n<d) and thus summarize the variability of a subset of analytes. Although they discard explicit dependencies with the remaining variables, they implicitly reflect the influence of the full joint distribution and therefore provide more robust information than univariate fits.

Conditionals

For two disjoint index sets A, B ⊂{1,…,d}, the conditional distribution of $X_{A}$ given $X_{B}=b$ is also Gaussian:(3)$$p(X_{A}|X_{B}=b)∼N(\mu_{A|B},\;\;\Sigma_{A|B})$$
where the partitioned mean and covariance are written as(4)$$X=\left[\begin{array}{c} X_{A}\\ X_{B} \end{array}\right],\;\mu =\left[\begin{array}{c} \mu_{A}\\ \mu_{B} \end{array}\right],\;\Sigma =\left[\begin{array}{cc} \Sigma_{AA} & \Sigma_{AB}\\ \Sigma_{BA} & \Sigma_{BB} \end{array}\right]$$
and the conditional parameters are(5)$$\mu_{A|B}=\mu_{A}+\;\Sigma_{AB}\;\Sigma_{BB}^{-1}\;\left(b-\mu_{B}\right)\;\Sigma_{A|B\;}=\;\Sigma_{AA}-\Sigma_{AB}\;\Sigma_{BB}^{-1}\;\Sigma_{BA}$$

Like marginals, conditional distributions reduce dimensionality by integrating out irrelevant coordinates. But first they restrict the distribution to a subset of axes under fixed values of another disjoint subset (m<d).

## 3. Results

### 3.1. Joint Distribution

We applied the proposed methodology to the metabolic biomarker dataset described in Section 2.1. As an initial step, we constructed the joint distribution of all lipid panel biomarkers together with relevant comorbidity markers. To normalize the skewed distributions, logarithmic transformations were applied CHOL, TRIG, HDL, LDL, and VLDL, while A1C and CREA were left untransformed. Outliers were generally retained to preserve the natural variability of the population, with the exception of physiologically implausible values, which were excluded using predefined biomarker-specific thresholds. An overview of the data preparation, model fitting, and inference workflows is provided in the Appendix A.

Subsequently, the BGMM was fitted to the full dataset, yielding the joint seven-dimensional distribution of all biomarkers, $p\left(X\right)$. The BGMM incorporates a Dirichlet process prior, which adaptively determines the effective number of active components. In this study, we specified a maximum of K = 7 Gaussian components, ensuring that several components could remain inactive if unsupported by the data.

The weights represent the relative proportion of the population assigned to each component. As shown in Figure 1, only two mixture components consistently carried substantial weight (>10%). The dominant component, corresponding to the healthy subpopulation, accounted for 40–80% of the distribution. Its weight was highest in younger age groups, where pathological conditions are less prevalent and the main distribution remains less distorted. The second major component captured pathological or outlier contributions, with its relative weight increasing in older age groups as comorbidities became more frequent. Together, these weights provide a quantitative view of how the balance between healthy and pathological subpopulations shifts across age and sex.

We optimized the model hyperparameters to bias the fit toward a dominant “central” Gaussian component representing the healthy population, while still allowing additional components to capture pathological or secondary subpopulations. The prior mean was estimated as the feature-wise median to reduce sensitivity to outliers, and the covariance prior was regularized with diagonal shrinkage to ensure stability and avoid ill-conditioning in higher dimensions. To encourage sparsity, the Dirichlet weight concentration prior was set to a very low value ($10^{-6}$), thereby limiting the number of active components. Conversely, the mean precision prior was set high (200.0) to constrain the component means more closely around the overall population mean.

The effect of the GMM is twofold: first, it separates multimodal Gaussian structures, and second, it effectively absorbs outliers in the distribution tails by allocating a specific component to them. In one dimension, the healthy and pathological contributions overlap substantially, so pathological values primarily shift the mean and increase the weight of the tail rather than forming distinct modes. In contrast, in the seven-dimensional space, pathological values appear as separate modes that collapse onto each other in the one-dimensional projection. Our results indicate the presence of two to three such satellite modes, each carrying a substantial portion of the total weight (Figure 1). As population segments become less healthy, the weight of these satellite modes increases at the expense of the central distribution, reflecting the growing influence of pathological subgroups.

### 3.2. Marginal Distributions and Reference Intervals

One-dimensional marginal distributions, $p\left(X_{i}\right)$, for each analyte were obtained from the joint distribution $p\left(X\right)$, as described in the Methods. From each marginal distribution, it is possible to calculate the limits corresponding to any quantile q. RIs were defined as the limits $\left(l,h\right)$ enclosing the central 95% of the distribution.

First, we estimated RIs for CHOL stratified by sex and age (Figure 2). The gray line represents the marginal distributions of CHOL as described in the Methods. During adolescence, CHOL RIs were similar in males and females. With increasing age, however, both the lower and upper limits rose, with a steeper slope in males. The values peaked around age 45 in males and 55 in females, after which both limits gradually declined. This decline is consistent with the effect of selective mortality, as discussed further in the Discussion.

The conditional distributions derived from the joint model provide deeper insight into the structure of RIs. In addition to the overall cohort RIs, we computed 95% intervals for sub-cohorts affected by comorbidities. In Figure 2, the red line corresponds to the diabetic sub-cohort (A1C = 9.0%) and the blue line to the renal disease sub-cohort (CREA = 2.0 mg/dL), where the values in of the diagnostic markers were chosen well in the unhealthy range. The presence of comorbidities markedly altered the lipid profile. Among younger adults with diabetes, the upper limit of CHOL was substantially elevated and the onset of selective mortality occurred earlier. Similarly, renal disease was associated with a pronounced increase in the upper limit of CHOL, although it did not seem to affect the selective mortality.

Conditional modeling also enabled the exclusion of comorbidity effects without requiring explicit diagnostic labels. By constraining analytes to healthy ranges (A1C = 5.7%, CREA = 1.0 mg/dL), the green line in Figure 2 captures the “true” RI for otherwise healthy adults. The near coincidence of the green (constrained) and gray (unconstrained) lines demonstrates that the BGMM successfully isolates the principal healthy distribution from pathological clusters. Nevertheless, along the comorbidity axes the distribution retains the imprint of disease effects, such that conditional probabilities in those directions still reveal how comorbidities distort lipid biomarkers. This illustrates that while the BGMM separates healthy and pathological populations, it also preserves meaningful covariance structure with comorbidity-related variables.

The RIs for the complementary cholesterol biomarkers, HDL and LDL, are shown in Figure 3 and Figure 4, respectively. HDL levels remained relatively stable across the cohort until middle age, after which a modest increase was observed, attributable to selective mortality (Figure 3). More striking, however, was the effect of comorbidities on HDL. In particular, diabetes caused a pronounced reduction in “good” cholesterol, underscoring its role as a strong contributor to cardiovascular risk [10].

The RIs for LDL (Figure 4) followed the same pattern as those for CHOL, since LDL values reported in clinical laboratories are typically not measured directly but calculated using the Friedewald formula: LDL=CHOL−HDL−VLDL where VLDL is estimated as VLDL=TRIG/5 [34]. As a result, LDL behavior largely mirrored that of total cholesterol. However, the impact of comorbidities was amplified, because LDL incorporates both the elevation of total cholesterol and the reduction in high-density lipoprotein (HDL, “good” cholesterol).

The RIs for TRIG are shown in Figure 5. Their pattern resembled that of CHOL, with values rising until middle age and then declining due to selective mortality. The influence of comorbidities, however, was distinct. Diabetes exerted an even stronger effect on TRIG than on CHOL, consistent with the close metabolic relationship between triglycerides and glucose. Renal disease, in contrast, showed its strongest impact in older individuals, significantly increasing triglyceride levels in the elderly population probably due to less efficient filtration.

The uncertainty in RI estimation was quantified using bootstrapping. Specifically, 100 random subsamples were drawn, each comprising 50% of the dataset, and RI limits were recalculated for each iteration. The final RI limits were taken as the mean across bootstrap samples, and the associated error was estimated as the standard deviation. The resulting errors were consistently small—on the order of a fraction of a percent—indicating high robustness of the estimates. Comparable stability was observed in our previous studies [27,29].

## 4. Discussion

### 4.1. RI Interpretation

Internationally accepted RIs for lipid biomarkers are largely derived from U.S. and European populations [6,7]. These conventional RIs are typically reported as fixed thresholds and are not stratified by sex, age, or race, despite well-documented physiological differences across demographic groups. In contrast, our approach uses real-world clinical laboratory data to derive population-specific RIs, as demonstrated here for the PR population. The scale and richness of the dataset further allow stratification by age and sex, yielding intervals that more accurately capture biological variation within the population.

A key observation from our results (Figure 2, Figure 3, Figure 4 and Figure 5) is the strong effect of age on lipid biomarker distributions. Applying a single RI across all ages fails to account for the normal, progressive decline in organ function, effectively treating age as a pathological state. In reality, most metabolic biomarkers—including those beyond the lipid panel—shift inexorably toward less favorable values with age [27,29]. Sex differences are also evident, driven by factors such as body size and hormone-related physiology. Thus, the conventional RIs used in current practice represent little more than population-wide averages, obscuring the biologically meaningful variation attributable to age and sex.

Our results for CHOL indicate that the upper RI limit in the PR population exceeds the widely recommended threshold of 200 mg/dL, even among young adults (Figure 2). Values rise quickly beyond the high-risk cutoff of 240 mg/dL, suggesting an accelerated trajectory toward dyslipidemia. A similar trend is observed for LDL, which largely mirrors total cholesterol, with upper limits exceeding the recommended threshold of 100 mg/dL (Figure 4). Triglycerides show the same pattern (Figure 5), with values generally above the established cutoff of 150 mg/dL. HDL levels are consistent with this unfavorable profile: the lower limit in our results falls below the conventional threshold of 40 mg/dL.

Together, these patterns suggest a population-wide shift toward higher cardiometabolic risk, likely influenced by dietary habits rich in fried foods and by the high prevalence of obesity, which increasingly affects younger age groups. Alternatively, this may suggest that the currently recommended cutoffs underestimate the physiological lipid ranges for the PR population—or perhaps for broader populations as well. It could also be that existing RIs for CHOL and LDL are set conservatively low, in part to encourage pharmacological intervention with statins. This interpretation aligns with recent evidence questioning whether elevated LDL levels are consistently associated with increased all-cause mortality [35].

It can also be argued that the distribution mean, rather than the conventional RI upper limit, is a more informative indicator of population health. The standard 95% interval is directly defined by the mean and standard deviation, $\left[\mu -1.96\sigma ,\;\mu +1.96\sigma \right]$, and narrower published RIs may simply reflect the limited diversity of the clinical trial cohorts from which they were derived. In contrast, our dataset captures the full PR population, providing a more representative picture. This underscores a broader issue with the way RIs and their associated abnormal flags are applied in practice: RIs define only hard cutoffs and do not convey how values are distributed within those limits. As a result, the laboratory abnormal flag is binary—indicating only whether a value lies inside or outside the interval. By modeling the full distribution, we can assign each individual measurement a percentile, offering a continuous measure of risk rather than a simple threshold. We have previously proposed this percentile-based framework as a more nuanced and clinically meaningful approach [27,29].

Finally, it is also important to emphasize the substantial sex differences in lipid profiles that are obscured by generalized recommendations. Females consistently exhibit lower CHOL and LDL levels, substantially higher HDL, and markedly lower TRIG compared to males—differences that are largely attributable to biological factors such as hormonal regulation and body composition. At the same time, the more rapid deterioration of lipid biomarkers observed in males likely reflects behavioral and lifestyle influences, compounding the biological baseline differences between the sexes.

### 4.2. Selective Mortality

The gradual rise in CHOL and LDL, coupled with a decline in HDL during early and mid-adulthood, aligns with the expected pattern of metabolic aging and physiological decline. This is consistent with the gradual loss of organ function due to normal wear and tear with age. Similar trajectories were seen in our earlier studies of CKD, where creatinine and urea levels increase steadily with age [27], and of CLD, where platelet counts and albumin concentrations progressively decrease [29].

However, lipid biomarkers display an unexpected pattern after midlife: CHOL and LDL values decline, while HDL rises. Although this could appear to reflect improved cardiovascular health, the paradox is better explained by selective mortality. Younger adults are resilient enough to survive despite adverse biomarker profiles, so their gradual health decline is visible in worsening trends. Beyond midlife, however, physiological reserves diminish, and the body can no longer compensate for chronic dysfunction. As a result, the least healthy individuals are more likely to die earlier, shifting the observed population distribution toward healthier values. The apparent improvement in biomarker profiles therefore reflects differential mortality rather than true physiological recovery. This interpretation is consistent with prior epidemiological studies reporting U-shaped cholesterol–mortality associations, where low cholesterol values in older adults were inversely associated with all-cause mortality and partly attributed to selective survival effects [36,37]. Nevertheless, similar reductions have also been observed longitudinally within individuals, suggesting that additional metabolic factors contribute [38].

The magnitude of selective mortality also varies across conditions. Some chronic dysfunctions, such as CKD, can often be tolerated until later in life and therefore do not substantially alter the overall trajectory of the lipid curves. In contrast, conditions that contribute more directly to early mortality, such as diabetes and cardiovascular disease, shift the onset of selective mortality to earlier ages. This difference is evident in our results (Figure 2, Figure 3, Figure 4 and Figure 5), where CKD does not substantially change the lipid curves, while DM produces a marked shift toward earlier onset of selective mortality.

For comparison, mortality rates for the PR population were estimated using U.S. Census data for the period 2020–2024, stratified by sex [39]. For each year, we calculated the fraction of individuals of age x who survived to age x+1 in the following year and then averaged these survival rates across years to reduce variability. The annual mortality rate, defined as (1−survival rate), is shown by age and sex in Figure 6a. Mortality remained relatively stable through early and middle adulthood, though subject to fluctuations likely attributable to migration and other demographic factors. Beginning in midlife, however, mortality rose exponentially, consistent with the Gompertz law, a well-documented demographic pattern of age-dependent mortality rise [40].

The corresponding cumulative survival function, $S\left(x\right)$, representing the fraction of the population surviving to age x, is shown in Figure 6b. Consistent with the RI curves, survival declines approximately exponentially at advanced ages, with males consistently exhibiting lower survival than females across the entire age range.

Although the survival function aggregates all causes of mortality and cannot be stratified by specific causes using the available census data, it nevertheless explains the patterns observed in the lipid panel RIs. Because male mortality is higher, individuals with adverse lipid profiles are removed from the cohort earlier, producing the earlier decline in elevated CHOL and LDL values in men. Similarly, sub-cohorts affected by specific chronic conditions would be expected to have steeper survival curves than the population average, leading to an earlier onset of mortality and corresponding shifts in their biomarker distributions.

### 4.3. Implication Network

To better understand how comorbidities influence lipid panel biomarkers, we constructed a comorbidity implication network. For this purpose, a set of diagnostic criteria was developed, as summarized in Table 2, and then joined biomarker data by patient ID and date to apply these rules. This framework allowed us to model disease co-occurrence and explore how multiple conditions interact within individuals. This methodology aligns with contemporary network medicine approaches—where diseases are represented as nodes and statistically derived associations as edges—to uncover patterns of comorbidity and aid in identifying preconditions and complications within patient trajectories [41,42].

After assigning diagnoses, we calculated the conditional probability that an individual has condition B given the presence of condition A(6)$$P\left(B|A\right)=\frac{N\left(A\cap B\right)}{N\left(A\right)}$$
where $N\left(A\cap B\right)$ is the number of individuals with both A and B, and $N\left(A\right)$ is the number of individuals with A. Together with the complementary probability $P\left(A|B\right)$, these values were used to construct the implication matrix, defined as $I\left[A,B\right]=P\left(A|B\right)$. This matrix serves as the foundation for the comorbidity implication network, where directed edges capture the strength and asymmetry of conditional relationships between diseases. Additional details on the construction of the implication matrix, together with its heatmap representation, are provided in Appendix A.

We then constructed a comorbidity network defined as a directed graph, using the implication matrix I as the adjacency matrix. The resulting network, illustrated in Figure 7, was analyzed with the Hyperlink-Induced Topic Search (HITS) algorithm to identify hub and authority nodes [43]. Conditions with high hub scores act as precursors, pointing toward multiple downstream comorbidities and serving as initiators of disease cascades. In contrast, conditions with high authority scores function as endpoints, receiving links from many hubs and representing common complications or outcomes of diverse pathological pathways.

The network provides a qualitative view of the relationships among comorbidities. As expected, both hypercholesterolemia and hypertriglyceridemia showed strong implications for CVD, reflected in both edge strengths and high hub scores (Figure 7a). Diabetes also emerged as an important precondition. Conversely, CVD appeared as the most common complication of hyperlipidemia (Figure 7b), while CKD was more closely associated as a complication of diabetes. The strong linkage between hyperlipidemia and CVD is partly tautological, since CVD risk is clinically estimated using cholesterol ratios (Table 2). Nevertheless, the network highlights important nuances: elevated triglycerides were more strongly associated with moderate CVD risk, whereas high cholesterol was more indicative of severe CVD risk. Furthermore, high cholesterol also appeared as a possible precondition for elevated triglycerides.

Even more importantly, the network helps to elucidate how comorbidities directly affect lipid biomarkers. For instance, diabetes appears as a precondition for elevated triglycerides. Impaired glucose metabolism in diabetes promotes hepatic triglyceride synthesis and reduces lipid clearance, leading to hypertriglyceridemia. This, in turn, amplifies cardiovascular risk by contributing to atherogenic dyslipidemia, characterized by high triglycerides, low HDL, and elevated small dense LDL particles.

### 4.4. Limitations and Future Work

#### 4.4.1. Ground Truth and Laboratory Data Limitations

The principal issue with underrepresented populations, such as PR, is that there are no cohort studies of population that could provide independent validation of the derived RIs. Moreover, chemical laboratory data represents only one side of the health profile of the individual. It lacks important invocators such as vital signs (height, weight, blood pressure, etc.) and other relevant information about the patients’ condition such as pregnancy, postpartum, or perimenopause. Better analyses would require linkage with electronic health records (EHRs) to enrich the laboratory data.

#### 4.4.2. Data Standardization and Harmonization

Although Abartys Health’s core business involves collecting, cleaning, and standardizing laboratory data across PR, challenges remain due to the lack of standardized and adherence to exchange formats in raw clinical data. Common issues that cannot be easily corrected include—incorrect or ambiguous coding of tests (that cannot distinguish between several closely related tests); lack of detailed metadata on analytical principles, reagents, and platforms that hamper data harmonization; lack of units or incorrect units, etc.

#### 4.4.3. Future Work

The application by this methodology is by no means specific to the PR population. It is fully generalizable provided that routine laboratory test results with demographic information (sex and age) are available. The additional features (e.g., ethnicity) can be easily incorporated to further stratify the RIs. In this case, the PR population serves not only as illustration but also provides valuable epidemiological information of an underrepresented population.

Furthermore, while here we only consider two comorbidities, this is not a limitation of the methodology. This is done to introduce and illustrate the idea of comorbidity networks in a simple form while capturing two of the major comorbidities—DM and CKD—that are highly prevalent in the population. Comorbidity networks construction from laboratory data is only limited by the possibility of diagnosing condition entirely based on laboratory data. There must be sufficient data for the principal biomarkers and the comorbidities available within the same time window.

## 5. Conclusions

This study demonstrates that Gaussian mixture modeling provides a robust framework for disentangling healthy from pathological distributions in large-scale, real-world laboratory data. By leveraging conditional probability, we were able to estimate definitive RIs for healthy cohorts without relying on diagnostic labels. Conditional modeling further revealed how comorbidities such as diabetes and renal disease systematically alter lipid distributions, while comorbidity implication networks clarified dependencies between conditions and highlighted key precursors and complications.

Unlike traditional RIs, which are reported only as fixed limits, our approach models the entire distribution of biomarker values in the population. This enables each individual result to be placed on a continuous percentile scale of risk, rather than being reduced to a binary abnormal flag. Such distribution-aware reporting provides a more nuanced basis for clinical decision-making.

Finally, our analysis highlights that the narrower RIs in published guidelines likely reflect the limited diversity of the clinical trial cohorts from which they were derived. In contrast, our framework captures the biological and demographic variability of the PR population, underscoring the need for population-specific standards derived from representative real-world data.

## Figures and Tables

**Figure 1 healthcare-13-02499-f001:**
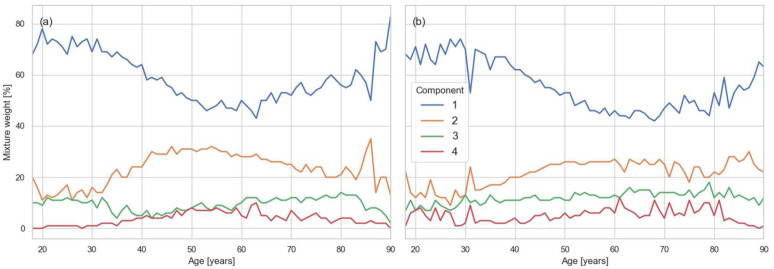
Weights (wₖ) of the major mixture components (1–4) of the lipid distributions identified by the BGMM, stratified by sex and age. Panel (**a**) shows results for males and panel (**b**) for females.

**Figure 2 healthcare-13-02499-f002:**
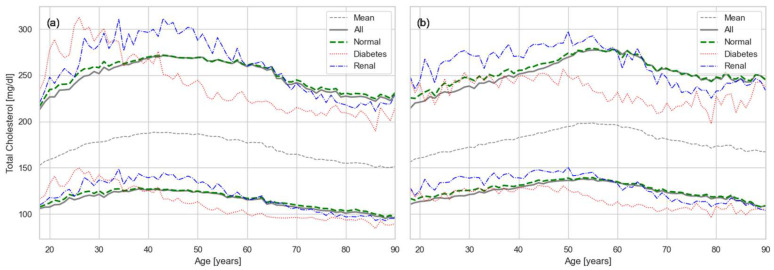
Age- and sex-specific RIs for CHOL: (**a**) males and (**b**) females. Solid gray lines show overall RIs, dashed gray show the mean of the distribution, dashed green show RIs restricted to metabolically normal individuals, while red and blue lines show conditional RIs for individuals with diabetes and renal disease, respectively.

**Figure 3 healthcare-13-02499-f003:**
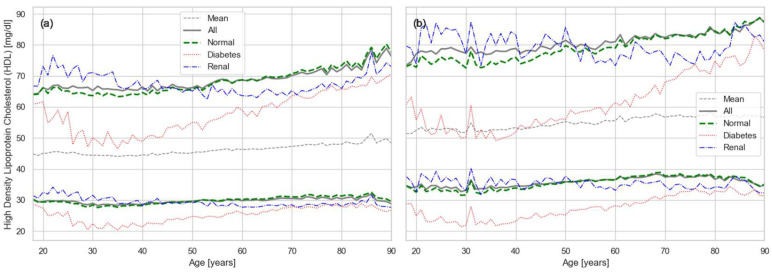
Age- and sex-specific RIs for HDL: (**a**) males and (**b**) females. Solid gray lines show overall RIs, dashed gray show mean of the distribution, dashed green represent RIs constrained to metabolically normal individuals, while red and blue lines indicate conditional RIs for diabetes and renal disease, respectively.

**Figure 4 healthcare-13-02499-f004:**
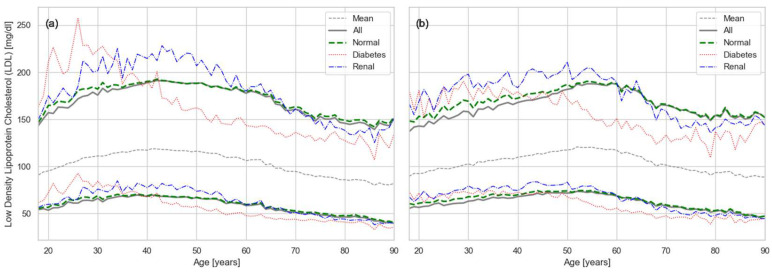
Age- and sex-specific RIs for LDL: (**a**) males and (**b**) females. Solid gray lines show overall RIs, dashed gray show the mean of the distribution, dashed green represent RIs constrained to metabolically normal individuals, while red and blue lines indicate conditional RIs for diabetes and renal disease, respectively.

**Figure 5 healthcare-13-02499-f005:**
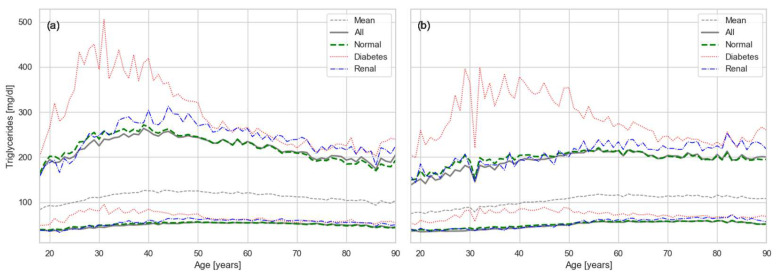
Age- and sex-specific RIs for TRIG: (**a**) males and (**b**) females. Solid green lines show overall RIs, dashed gray show the mean of the distribution, dashed green represent RIs constrained to metabolically normal individuals, while red and blue lines indicate conditional RIs for diabetes and renal disease, respectively.

**Figure 6 healthcare-13-02499-f006:**
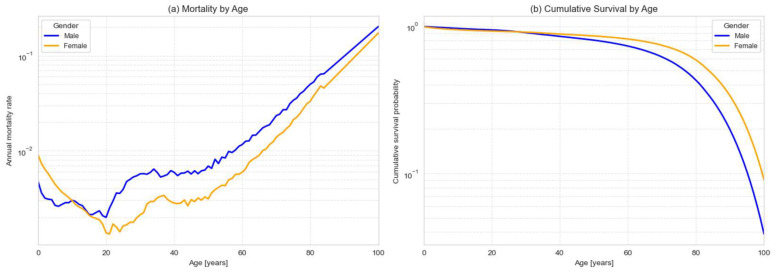
Mortality and survival functions for the PR population (2020–2024), stratified by sex. (**a**) Annual mortality rates by age, shown on a logarithmic scale. (**b**) Corresponding cumulative survival functions.

**Figure 7 healthcare-13-02499-f007:**
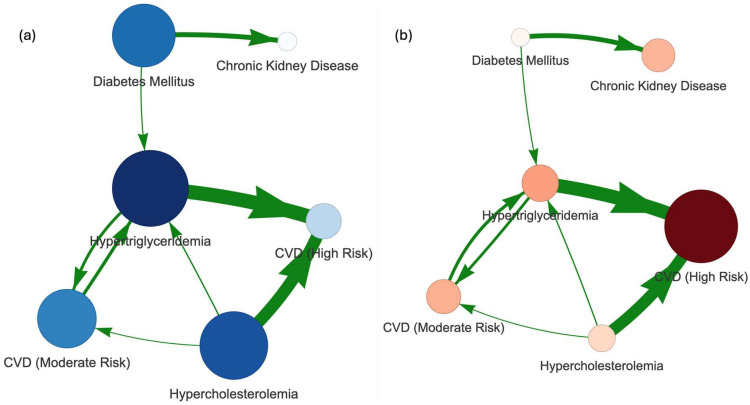
Comorbidity implication network for cardiovascular risk conditions. (**a**) Hub representation, where larger nodes indicate conditions with strong hub scores (precursors leading to multiple downstream comorbidities). (**b**) Authority representation, where larger nodes correspond to conditions with high authority scores (common complications influenced by multiple precursors). Arrow direction and thickness indicate conditional probabilities P(A|B), and only the strongest 75% of edges are displayed for clarity.

**Table 1 healthcare-13-02499-t001:** Lipid panel and comorbidity biomarkers: number of test results and unique individuals for the period 2019–2024. The merged (“Joined”) dataset refers to the subset of cases where all biomarkers were available in the same time window.

Biomarker	Results	Persons	Male	Female
Total cholesterol (CHOL) Triglycerides (TRIG) Low-density lipoprotein (LDL) High-density lipoprotein (HDL)	4,349,050	1,353,928	574,275	779,653
Hemoglobin A1c (A1C)	2,322,403	842,429	347,790	494,639
Creatinine (CREA)	6,003,119	1,613,033	689,537	923,496
Joined	1,775,134	717,312	296,470	420,842

**Table 2 healthcare-13-02499-t002:** Diagnostic criteria used to define cardiovascular conditions and comorbidities based on laboratory biomarkers. Thresholds were selected from established clinical guidelines to identify hyperlipidemia, CVD (moderate and high risk), DM, and CKD.

Biomarker	Results
Hypercholesterolemia	cholesterol > 200
Hypertriglyceridemia	triglycerides > 150
CVD (High Risk)	(cholesterol/hdl > 5.0) or (ldl/hdl > 3.5)
CVD (Moderate Risk)	(4.0 < cholesterol/hdl < 5.0) or (2.5 < ldl/hdl <3.5)
Diabetes Mellitus	hga1c > 6.5
Chronic Kidney Disease	creatinine > 1.3

## Data Availability

The data that support the findings of this study were licensed from Abartys Health for the purposes of this study alone and are not publicly available. Data access requests can be addressed to the corresponding author who would relay them to Abartys Health. Reasonable requests for access to the original code used to analyze the data can be directed to the corresponding author.

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
