# Peer review of "Modeling Multivariate Distributions of Lipid Panel Biomarkers for Reference Interval Estimation and Comorbidity Analysis"

_healthcare, 2025, doi:10.3390/healthcare13192499_

Round 1
Reviewer 1 Report
Comments and Suggestions for Authors
The manuscript relates a data-driven method to derive population-specific reference intervals for lipid biomarkers using large-scale real-world laboratory data. It highlights how Gaussian mixture modeling can separate healthy/pathological subgroups and account for age, sex, and comorbidities to improve clinical interpretation.
Comments, regarding the introduction section, the provided data are not well directed to the main problem. It is necessary to balance the information regarding the pathologies that use the lipid panel to direct these data to the reference interval.
Lines 96 to 99 should not be part of the introduction; this is part of the methods
The mention in these lines that the RI was estimated for chronic kidney disease and chronic liver disease, but the last one was not mentioned before.
It is necessary to improve the introduction section because, as it is articulated is not well understood
The paragraph contained between lines 100 and 111 should be part of the methods section.
Statement contained in lines 103-104 should have a reference, please.
In the materials and method section
The comprehensive metabolic panel (CMP) is introduced, but this should be included in the introduction section, as well as the CREA and A1C concepts; otherwise, it is not connected and is difficult to get the proper message before the methods section (lines 126 to 131).
It is impressive the number of numbers you have in this study.
Table 1, since data of CREA and A1C is not introduced before, here is not clear why you used. In addition, joined is not well described; please add a phrase in the manuscript describing the concept.
In the results section it is not clear why the weight is included in the text, and so is the survival rate, please add a brief explanation. Survival was not described at the beginning; therefore, it is a new component that should be described in the introduction section.
Figure 1 does not explain what a and b are in the graphs shown, nor the components from 1 to 4.
In the discussion, Figure 1 is not discussed; please add a word related to it.
Discussion content between lines 364 and 371 is related to Figure 1? Maybe direct the information towards this figure.
Since sex differences are discussed in the last paragraph of the discussion, did you conduct further analysis regarding lipids during women's pregnancy, postpartum, and perimenopause? This is important to visualize, since it is well known that during these periods, lipid levels change.
Section 4.2 should be included in the abstract a well.
Section 4.3 should be highlighted in the abstract as well, these data are important as a contribution to the field is this specific population, and point out what is the difference in comparison to the data published in the guides from EE.UU and Europe.
In general, it is necessary to connect all the elements mentioned in the results section to the introduction to ensure a clear understanding of the manuscript. The math presented in the methodology can be supplementary information, as it is currently presented, complicates the lecture. It is important to highlight the differences you found in the general numbers of the biomarkers analyzed and also the cardiometabolic risk calculation. This is not present in your abstract
Author Response
Comment 1. Introduction.
- The provided data are not well directed to the main problem. It is necessary to balance the information regarding the pathologies that use the lipid panel to direct these data to the reference interval.
- Lines 96 to 99 should not be part of the introduction; this is part of the methods
- The mention in these lines that the RI was estimated for chronic kidney disease and chronic liver disease, but the last one was not mentioned before.
- It is necessary to improve the introduction section because, as it is articulated is not well understood
- The paragraph contained between lines 100 and 111 should be part of the methods section.
- Statement contained in lines 103-104 should have a reference, please.
Response 1. We thank the reviewer for these helpful comments. In the revised manuscript, we have substantially restructured the Introduction to improve clarity and flow. The section begins with the burden of cardiovascular disease, then introduces the lipid panel as the main tool for cardiovascular risk assessment. We next highlight the impact of comorbidities such as diabetes and chronic kidney disease on lipid distributions, and introduce the comorbidity-related biomarkers (A1C and CREA) included in the CMP. The concept of reference intervals is then introduced, together with the limitations of traditional methods. Finally, we summarize our previous work and clearly state the novelty of the present study. To avoid confusion, methodological details that previously appeared in lines 96–99 and 100–111 have been moved or rephrased. Chronic liver disease is now mentioned explicitly alongside chronic kidney disease in the Introduction, and we have added the appropriate references. We believe that the revised structure ensures that the clinical context, methodological rationale, and contributions of the present study are presented in a balanced and connected manner.
Comment 2. Materials and Methods.
- The comprehensive metabolic panel (CMP) is introduced, but this should be included in the introduction section, as well as the CREA and A1C concepts; otherwise, it is not connected and is difficult to get the proper message before the methods section (lines 126 to 131).
- Table 1, since data of CREA and A1C is not introduced before, here is not clear why you used. In addition, joined is not well described; please add a phrase in the manuscript describing the concept.
Response. As noted above, A1C and CREA are now introduced in the Introduction, where their clinical relevance and connection to the lipid panel are explained. In addition, we have clarified the meaning of “joined” in the Table 1 caption, specifying that it refers to the merged dataset combining lipid panel, A1C, and CREA results for the same individuals. The meaning of the “joint” dataset is also discussed in the paragraph below the table.
- In the results section it is not clear why the weight is included in the text, and so is the survival rate, please add a brief explanation. Survival was not described at the beginning; therefore, it is a new component that should be described in the introduction section.
- Figure 1 does not explain what a and b are in the graphs shown, nor the components from 1 to 4.
Response. In the revised manuscript, we have clarified the role of the mixture component weights, both in the text and in the Figure 1 caption, and explained that they reflect the relative proportion of healthy versus pathological subpopulations across age and sex.
Regarding survival, we agree that this concept arises only after observing the counterintuitive biomarker patterns in the Results. For this reason, it is not logical to introduce it in detail in the Introduction. Instead, we now mention it briefly in the novelty statement at the end of the Introduction, and then explain it fully in the Results and Discussion, where it is first observed and interpreted.
Comment 3. Discussion
- Figure 1 is not discussed; please add a word related to it.
- Discussion content between lines 364 and 371 is related to Figure 1? Maybe direct the information towards this figure.
Response. As noted above, we have improved the caption of Figure 1 to define the panels and components clearly. In addition, we added a dedicated paragraph immediately following the figure in the Results section that discusses the meaning of the component weights and their variation by age and sex. This new text explicitly references Figure 1 and explains how the mixture components reflect the balance between healthy and pathological subpopulations.
- Since sex differences are discussed in the last paragraph of the discussion, did you conduct further analysis regarding lipids during women's pregnancy, postpartum, and perimenopause? This is important to visualize, since it is well known that during these periods, lipid levels change.
Response. We thank the reviewer for raising this important point. Unfortunately, pregnancy status and reproductive history are not directly available in the laboratory results we analyzed. Such analyses would require linkage with detailed electronic health records (EHRs), which was beyond the scope of the present study. We have therefore added this as a limitation in the revised manuscript.
- Section 4.2 should be included in the abstract a well.
- Section 4.3 should be highlighted in the abstract as well, these data are important as a contribution to the field is this specific population and point out what is the difference in comparison to the data published in the guides from EE.UU and Europe.
Response. We thank the reviewer for this encouraging comment. In the revised manuscript, we have updated both the Abstract and the Introduction to highlight these additional contributions. The Abstract now mentions that selective mortality patterns were examined to explain counterintuitive age trends in lipid values and that comorbidity implication networks were constructed to capture interdependencies among conditions. Similarly, the final paragraph of the Introduction has been expanded to list these two aspects as further novelties of the present study. These changes ensure that the key findings and contributions of sections 4.2 and 4.3 are clearly introduced at the outset.
Reviewer 2 Report
Comments and Suggestions for Authors
Dear Authors,
We appreciate your submission to the journal of healthcare. The study addresses an important subject, but there are a few aspects that need improvement and clarity, including the methodology detail, clinical applicability, and result confirmation. Please respond in detail to the following major and minor comments. Following extensive revision, a revised version will be reviewed.
Major Comments
- The manuscript offers a viable method for comorbidity analysis and reference interval (RI) calculation based on Bayesian Gaussian Mixture Models. It would be better to specify the innovation in relation to the authors' earlier works (refs. 27–29). Please draw attention to what is actually novel in this case.
- There is no external validation of the methodology against established RI databases or independent clinical cohorts. Even a small amount of validation would greatly support the findings.
- Puerto Rican populations are the subject of the analysis. The topic of generalisability to different populations and whether demographic variations restrict wider applicability should be covered in more detail.
- Although the concept of selective mortality is intriguing, it needs more robust epidemiological or statistical support than descriptive patterns.
- More information is required regarding the selection of modifications for each biomarker and the identification and elimination of outliers. To ensure reproducibility, this is crucial.
- Although Figures 2–5 are the main focus, the captions are long and a little repetitive. Please make sure that captions are clear and concise, and think about adding RI summary tables.
- The comorbidity implication network is immature despite being creative. Could the authors give clinical interpretations and more precise quantifications of edge strengths (such conditional probabilities)?
- Although the argument that percentile-based risk estimations are easier to interpret than binary RI flags is strong, the text might benefit from a real-world or hypothetical case study to illustrate how they are used in clinical decision-making.
- The constraints are recognized, but more information is required about the difficulties in harmonizing data (several labs, test procedures, and units) and the possible biases in the resulting RIs.
- Despite being well-written overall, the introduction and discussion contain some verbose passages. Healthcare professionals would find it easier to read a presentation that is more succinct.
Minor Comments
- Please make sure that all acronyms (such as BGMM and CMP) are defined in both the abstract and the text before they are used.
- Explicitly stating the gender distribution and percentages of missing data would enhance Table 1.
- Throughout the manuscript, the terms "RI" and "reference interval" should be used consistently.
- Please confirm the 2025 access dates listed in several references (such as CDC web sites), as they appear to be inconsistent.
- Verify that all text, figures, and tables use the same units (e.g., mg/dL, %).
- Figure quality should be enhanced for publishing, specifically the legibility and resolution of the lines in Figures 2–5.
- Some references (like Ref 39) have HTML symbols that need to be fixed, like "\&c."
- A flow diagram that summarizes the workflow for data pretreatment and analysis would be helpful in the Methods section.
- To make it more succinct without sacrificing clarity, the word count could be somewhat decreased in the abstract.
- "comorbidity implication network" and "condition implication network" are interchangeable terms. Standardize the terms.
Author Response
Comment 1. The manuscript offers a viable method for comorbidity analysis and reference interval (RI) calculation based on Bayesian Gaussian Mixture Models. It would be better to specify the innovation in relation to the authors' earlier works (refs. 27–29). Please draw attention to what is actually novel in this case.
Response. In the revised manuscript, we have clarified the novelty of the present study in relation to our earlier work. In particular, the final paragraph of the Introduction has been rewritten to emphasize that this study (i) extends the BGMM framework to multivariate data across the entire lipid panel, (ii) applies conditional probability distributions to better account for comorbidity effects without diagnostic codes, (iii) identifies selective mortality patterns that explain counterintuitive biomarker trends at older ages, and (iv) introduces comorbidity implication networks to characterize interdependencies among conditions. We have also revised the Abstract to reflect these contributions more explicitly, ensuring that the innovations are clear from the outset.
Comment 2. There is no external validation of the methodology against established RI databases or independent clinical cohorts. Even a small amount of validation would greatly support the findings.
Response. We acknowledge this limitation. Unfortunately, we are not aware of any cohort studies of the Puerto Rican population that could serve as an external validation dataset. We have added this point explicitly to the Limitations section. At the same time, we believe this highlights the strength of the present work, as it provides the first comprehensive study of this type for the Puerto Rican population, filling an important gap in the literature.
Comment 3. Puerto Rican populations are the subject of the analysis. The topic of generalizability to different populations and whether demographic variations restrict wider applicability should be covered in more detail.
Response. We have substantially expanded the Limitations section to clarify that while the present results are specific to the Puerto Rican population, the methodology itself is general and can be directly applied to any population with sufficient laboratory data and basic demographic information (sex and age).
Comment 4. Although the concept of selective mortality is intriguing, it needs more robust epidemiological or statistical support than descriptive patterns.
Response. We have revised the selective mortality section to better connect it to the RI results and added a couple of references that support our hypothesis. We have also tried a few methods to use the survival function to resample the results in such a way as to recover the distribution of the results if the individuals were not taken out of the sample due to mortality. Unfortunately, this proved to be challenging and the results not convulsive, therefore, it remains in the scope of future work. Nevertheless, we found it important to discuss the hypothesis.
Comment 5. More information is required regarding the selection of modifications for each biomarker and the identification and elimination of outliers. To ensure reproducibility, this is crucial.
Response. The choice of transformations for each biomarker is described in the Methods and Results sections, and the outlier removal procedure is also outlined in the Results.
Comment 6. Although Figures 2–5 are the main focus, the captions are long and a little repetitive. Please make sure that captions are clear and concise, and think about adding RI summary tables.
Response. We have revised the figure captions to make them more concise while keeping them clear and self-contained.
Comment 7. The comorbidity implication network is immature despite being creative. Could the authors give clinical interpretations and more precise quantifications of edge strengths (such conditional probabilities)?
Response. We have expanded the Supplement with additional discussion of the comorbidity implication network, including clinical interpretation of key associations. To provide more precise quantification of edge strengths, we now include the implication matrix as a heatmap.
Comment 8. Although the argument that percentile-based risk estimations are easier to interpret than binary RI flags is strong, the text might benefit from a real-world or hypothetical case study to illustrate how they are used in clinical decision-making.
Response. Similar examples have been provided in our previous work, to which we also refer the reader for further illustration.
Comments 9. The constraints are recognized, but more information is required about the difficulties in harmonizing data (several labs, test procedures, and units) and the possible biases in the resulting RIs.
Response. We have substantially expanded the Limitations section to outline the common issues with real-world data. While best effort was made to harmonize correctly the tests across different labs and to clean and convert units, the improvement of the dataset is beyond the scope and purpose of the paper. This is the core business and know-how of Abartys Health, the company that provided the data.
Comment 10. Despite being well-written overall, the introduction and discussion contain some verbose passages. Healthcare professionals would find it easier to read a presentation that is more succinct.
Response. In the revised manuscript, we have shortened and improved both the Introduction and the Discussion, as outlined in our responses to the comments above. These revisions streamline the presentation, reduce repetition, and make the flow of arguments clearer.
Minor issues.
- Please make sure that all acronyms (such as BGMM and CMP) are defined in both the abstract and the text before they are used.
Response. Fixed.
- Explicitly stating the gender distribution and percentages of missing data would enhance Table 1.
Response. Added.
- Throughout the manuscript, the terms "RI" and "reference interval" should be used consistently.
Response. Corrected.
- Please confirm the 2025 access dates listed in several references (such as CDC web sites), as they appear to be inconsistent.
Response. Corrected.
- Verify that all text, figures, and tables use the same units (e.g., mg/dL, %).
- Figure quality should be enhanced for publishing, specifically the legibility and resolution of the lines in Figures 2–5.
Response. Corrected.
- Some references (like Ref 39) have HTML symbols that need to be fixed, like "\&c."
Response. We use a citation manager, and those issues reflect issues with the encoding of the database there the information is fetched from. We went through the references to manually correct any obvious issues.
- A flow diagram that summarizes the workflow for data pretreatment and analysis would be helpful in the Methods section.
Response. We thank the reviewer for this suggestion. A flow diagram has been added to the Supplement (Supp. Figs. 1–2), summarizing the workflows for data preparation, model fitting, and inference.
- To make it more succinct without sacrificing clarity, the word count could be somewhat decreased in the abstract.
Response. The revised abstract is kept within the 250-word limit with adding additional text about the selective mortality and comorbidity implication networks.
- "comorbidity implication network" and "condition implication network" are interchangeable terms. Standardize the terms.
Response. Standardized to comorbidity implication network in the revised version.
Reviewer 3 Report
Comments and Suggestions for Authors
The manuscript is clear and systematic in describing its methods, explaining why Bayesian Gaussian Mixture Models were used, how outliers were handled, and how hyperparameters were chosen, which makes the work reproducible. Overall, the study is novel because it proposes percentile-based risk assessment instead of simple “normal/abnormal” thresholds, offering a more meaningful approach for clinical use.
However, addressing the following points could further strengthen the manuscript:
- Limited Comorbidity Scope:The study only analyzes diabetes and chronic kidney disease. Other common conditions like hypertension or obesity are not included, which limits the completeness of the comorbidity network.
- Lack of Independent Validation:The reference intervals are based solely on internal data from Puerto Rican clinical labs. No external cohort is available for validation, so the generalizability of the results is uncertain.
- Data Quality Issues:Laboratory data had inconsistencies in test codes, units, and formats, which could introduce small biases in modeling or outlier detection.
- Modeling Assumptions:Hyperparameters were tuned to favor a “healthy” central component, which might underrepresent multimodal patterns in sicker subgroups. No sensitivity analyses were performed for diagnostic thresholds or modeling choices.
- Population Representation:The dataset reflects individuals who access healthcare and may overrepresent people with health issues, potentially skewing results away from truly healthy populations.
- Clinical Interpretation:The link to real-world clinical decision-making could be stronger. More explicit discussion on how these findings might affect risk assessment or treatment thresholds (e.g., statin use) would improve translational value.
Author Response
Comment 1. Limited Comorbidity Scope: The study only analyzes diabetes and chronic kidney disease. Other common conditions like hypertension or obesity are not included, which limits the completeness of the comorbidity network.
Response. We thank the reviewer for the positive feedback. As noted in the revised Limitations, our comorbidity analysis was restricted to diabetes and CKD to illustrate the methodology while keeping the focus on the primary objective of estimating RIs. Data on hypertension and obesity are not included in routine laboratory results and would require additional sources. The framework, however, is fully generalizable and can be extended to other conditions when suitable data are available.
Comment 2. Lack of Independent Validation: The reference intervals are based solely on internal data from Puerto Rican clinical labs. No external cohort is available for validation, so the generalizability of the results is uncertain.
Response. As noted also in response to Reviewer 2, no external cohort data are available for the Puerto Rican population, and we have highlighted this as a limitation.
Comment 3. Data Quality Issues: Laboratory data had inconsistencies in test codes, units, and formats, which could introduce small biases in modeling or outlier detection.
Response. As noted, inconsistencies in laboratory test codes, units, and formats were added to the Limitations, where we acknowledge the potential for small biases in modeling and highlight the need for broader standardization efforts.
Comment 4. Modeling Assumptions: Hyperparameters were tuned to favor a “healthy” central component, which might underrepresent multimodal patterns in sicker subgroups. No sensitivity analyses were performed for diagnostic thresholds or modeling choices.
Response. We thank the reviewer for raising this point. Based on our experience, biomarker distributions in one dimension generally do not exhibit multiple modes, but rather heavy tails. In the multivariate setting, however, the model does identify separate modes, as illustrated in Figure 1. Hyperparameters were tuned to favor the central “healthy” component, but we acknowledge that without ground truth data, a full sensitivity analysis of thresholds and modeling choices cannot be performed. This has been noted in the Limitations.
Comment 5. Population Representation: The dataset reflects individuals who access healthcare and may overrepresent people with health issues, potentially skewing results away from truly healthy populations.
Response. We agree with the reviewer that clinical laboratory data may overrepresent individuals with health issues, since healthier people tend to seek care less frequently. This is a general limitation of real-world data and has been noted in the Limitations. We also note that this bias diminishes with age, as healthcare utilization becomes more common in older populations.
Comment 6. Clinical Interpretation: The link to real-world clinical decision-making could be stronger. More explicit discussion on how these findings might affect risk assessment or treatment thresholds (e.g., statin use) would improve translational value.
Response. As mentioned above, we refer the reader to our previous work, where similar translational implications were illustrated.